# Eliminating mother-to-child transmission of HIV in Tanzania calls for efforts to address factors associated with a low confirmatory test

**Baraka M. Morris**[1]*, **Mukome Nyamhagata**[2], **Edith Tarimo**[1], **Bruno Sunguya**[1]

**1** Muhimbili University of Health and Allied Sciences, Dar es salaam, Tanzania, **2** Ministry of Health, Dodoma, Tanzania

* bamalaki@yahoo.com

**Data Availability Statement:** The custodian of patients' level data used in this study is the Ministry of Health, Tanzania. For research purposes, data

## Abstract

Option B+ approach for prevention of mother-to-child transmission (PMTCT) has demonstrated the potential to eliminate pediatric HIV infections. Its success depends on early infant diagnosis (EID) of HIV among the exposed infants within the first 6 weeks, and a subsequent confirmatory HIV test within 18 months. However, most mothers enrolling in option B+ in Tanzania do not come for such confirmatory tests. We examined factors associated with the turning-up of mother-baby pairs on the PMTCT program for a confirmatory HIV testing 18 months post-delivery in Tanzania. This study utilized longitudinal data collected between 2015 and 2017, from 751 mother-baby pairs enrolled in the PMTCT-option B+ approach in 79 health facilities from the 12 regions of Tanzania-mainland. Only 44.2% of 751 mother-baby records observed received the HIV confirmatory test by the 18th month. Mothers aged 25 years or above (adults' mothers) were 1.44 more likely to turn up for confirmatory HIV testing than young mothers; mothers with partners tested for HIV were 1.74 more likely to have confirmatory HIV testing compared with partners not tested for HIV. Newly diagnosed HIV-positive mothers were 28% less likely to bring their babies for a confirmatory HIV-testing compared to known HIV-positive mothers. Mothers with treatment supporters were 1.58 more likely to receive confirmatory HIV-testing compared to mothers without one. Mother-baby pairs who collected DBS-PCR-1 were 3.61 more likely to have confirmatory HIV-testing than those who didn't collect DBS-PCR-1. In conclusion, the confirmatory HIV testing within 18 months among mother-baby pairs enrolled in the Option B+ approach is still low in Tanzania. This is associated with low maternal age, having a male partner not tested for HIV, lack of experience with HIV services, lack of treatment supporters, and failure to take the DBS-PCR-1 HIV test within the first two months post-delivery.

## Introduction

The overall risk of HIV transmission from mother-to-child without any intervention ranges from 20% to 45% [1–3]. Such unprecedented risk can be ameliorated with a successful implementation of preventing mother-to-child transmission (PMTCT) interventions that includes screening and treatment of both mothers and their newborn [4–6]. Remarkable efforts have

are usually made available upon a reasonable request to the Permanent Secretary of the Ministry of Health. Data sharing with the third-party requires a signed Data Transfer Agreement provided by the National Health Research and Ethics Committee and a request to the Permanent Secretary of the Ministry of Health when the requirements are fulfilled. Our permission was not extended to such data sharing. Therefore, if these data will be required and the national requirements mentioned above have been fulfilled, anyone will be able to obtain the data set in the same manner that I had obtained it. Permanent Secretary, Ministry of Health P.O Box 743, Dodoma Telephone: +255-26-2323267/5 Mobile: +255-26-2342000/5 Email: ps@afya.go.tz Link: https://www.moh.go.tz/.

**Funding:** This work was supported by the Afya Bora Consortium Fellowship, which is funded by the U.S. President's Emergency Plan for AIDS Relief (PEPFAR) through funding to the University of Washington's International AIDS Education and Training Center (IAETC) under cooperative agreement U91 HA06801 from the Health Resources and Services Administration (HRSA) Global HIV/AIDS Bureau. The funders had no role in study design, data collection and analysis, decision to publish, or preparation of the manuscript.

**Competing interests:** The authors have declared that no competing interests exist.

been made in scaling up PMTCT services globally. Such efforts have resulted in marked improvement in PMTCT among countries with a heavy burden of HIV/AIDS [7]. Nevertheless, despite marked efforts and investment, the unprecedented burden of new infections still occurs among HIV-exposed newborns globally—Tanzania is no exception.

The World Health Organization (WHO) recommended the adoption of universal antiretroviral treatment or Option B+, as the preferred PMTCT approach in LMICs with high HIV prevalence, high fertility, and extended breastfeeding [5, 8]. Under this plan, all women who test HIV positive are enrolled in the lifelong ART soon after HIV diagnosis, regardless of their CD4 count, or HIV clinical stage. The implementation of the Option B+ program requires women and their infants to access to PMTCT cascade of HIV prevention services for two years post-delivery [9]. These services aim to ensure early infant diagnosis of HIV(EID) among HIV-exposed children and timely initiation of treatment once they are diagnosed with HIV. To determine whether these babies are infected, mother-baby pairs undergo a series of HIV testing. The WHO and Tanzania National Guidelines for the Management of HIV and AIDS of 2019 recommend the first test to be taken before the baby is six weeks of age and the final confirmatory HIV test to be taken at 18 months of age [10, 11]. If the child is infected, both the mother and her child are transferred to Care Treatment Clinics (CTC) [8, 9, 12].

Adopting Option B+ in Tanzania still faces several challenges similar to many other sub-Saharan countries. Studies reveal that a significant proportion of women decline the PMTCT service outright, while another group silently defaults from care [13, 14]. The proportion of infants who receive confirmatory HIV tests is also very low. Only 34.6% of babies from HIV-infected mothers are brought for confirmatory HIV testing at 18 months of age [15]. Low turn-up for confirmatory HIV testing denies the opportunity for early diagnosis and treatment among children who might have been infected during breastfeeding. Findings show that most of these undiagnosed children are likely to die before their second birthday if they do not receive treatment [5, 12, 16].

Evidence on the reasons and causes for low turn-up for confirmatory tests remains inconclusive and varies in Tanzania like in other countries. Thus, this study investigated factors associated with poor turn-up for confirmatory HIV testing by the 18th month of age among mother-baby pairs enrolled in the PMTCT Option B+ cascade in Tanzania.

## Materials and methods

### Study design and population

This secondary analysis study used data from a cohort of mother-baby pairs who received PMTCT care in 79 health facilities in 12 selected regions of Tanzania between 2015–2017. Tanzania approved the National Guidelines for the implementation of Option B+ in 2013. According to this guideline, all pregnant mothers who tested HIV positive were counseled and enrolled in the Option B+ approach where they were monitored from their first antenatal clinic visit until their babies were two years old [5, 17]. Data was collected to monitor and evaluate the adherence to treatment among mother-baby pairs enrolled in the Option B + approach of PMTCT.

### Sample size and sampling

Multistage random sampling was used to select 79 health facilities in the 12 regions of Tanzania mainland for the primary study. In these health facilities, a total of 4,738 HIV-positive mother-baby pairs were enrolled in the Option B+ program from 2015 to 2017. In the current study, only 751 HIV-positive mother-baby pairs were analyzed owing to the completeness of their medical records.

## Variables and measures

The outcome variable was the infant's confirmed HIV status within the 18th month of age. The outcome responses were categorical and measured using a nominal scale (dichotomous -Yes/No), where "Yes" was for mother-baby pairs who turned in for confirmatory HIV testing and "No" was for mother-baby pairs who did not turn for confirmatory HIV testing (Lost to Follow up). All babies who tested HIV-positive in the first dry blood smear (DBS-PCR-1) were included among those who received the confirmatory HIV test. This did not include babies who died in the course of 18 months.

Independent variables included eight variables. These are: Maternal age groups defined based on the WHO age groups where mothers aged 24 years and less form the young group and those aged 25 years and above from the adult group (2) Gravidity defined as the number of times that a woman has been pregnant. Gravidity was classified into two categories, the first category for mothers in their first & second pregnancies (G1-G2) and the second category for mothers with more than two pregnancies (G3 –max). (3) Gestational age is the measure (usually in weeks) of the age of a pregnancy. Normally, the gestation age is grouped into three groups (trimesters). However, the WHO recommends women start antenatal care in their first trimester. In this study, the gestation age was grouped into two categories, the first category for mothers who were enrolled in their 1st trimesters and the second category for mothers who enrolled in the 2nd and 3rd trimesters. (4) Live with Partner (marital status) measured as a nominal categorical (dichotomous -Yes/No) variable where "Yes" was for married and cohabited mothers and "No" was for Single, divorced, and widowed mothers. (5) Partner tested measured as a categorical variable using a nominal scale (dichotomous -Yes/No), where "Yes" was for mothers with a partner who has tested for HIV and "No" for mothers with a partner who has not tested for HIV (6) Maternal HIV status at enrolment measured as a nominal categorical variable with two groups. The first group was for mothers whose HIV status was already known as HIV positive and the second group was for mothers who were newly diagnosed as HIV positive at enrollment in antenatal care (Known HIV +ve and New HIV +ve). (7) Presence of treatment supporters was measured dichotomously using a nominal scale. "Yes" was for mothers with treatment supporter(s) and "No" was for mothers without treatment supporter(s). (8) Lastly, Collection of the first dry blood smear (DBS-PCR-1) within the first two months post-delivery. The WHO and the Tanzania National Guidelines for the Management of HIV and AIDS of 2019 recommend that HIV-exposed infants have their first HIV test within 6 weeks of age for early infant diagnosis of HIV (EID). However, during the data collection, the tool used for data collection used the guideline of that time that required mother-baby pairs enrolled on Option B+ PMTCT cascade to have their first HIV test within 2 months of age. For each mother-baby pair it was recorded whether DBS-PCR-1 was collected or not and hence, was measured dichotomously (Yes/No) using a nominal scale.

## Loss to follow-up

Loss to follow-up was one of the child's final status at the last visit. Likewise, this was one of the observations for not showing up for confirmatory HIV testing at the 18th month age of the baby. However, in this study, we did not analyze the child's last status, because child status is determined at the end of two years.

## Data analysis

The analysis focused on examining the upshots of Option B+ on the compliance to services among mother-baby pairs enrolled in the PMTCT program in Tanzania mainland. Factors associated with poor turn-up for confirmatory HIV testing within 18 months of infant age

among these mother-baby pairs were analyzed as well. These factors were placed into four groups namely: maternal social-demographic characteristics (maternal age, gravidity, gestation age at enrolment, and living with a partner), experience with HIV services on enrolment (maternal HIV status), partner involvement on turn-up for confirmatory HIV testing (Partner tested), and presence of treatment supporters (treatment supporter). The analysis examined whether these factors were associated with turn-up for confirmatory HIV testing.

Data were cleaned and analyzed using the STATA version 15[th] software package. Measures of central tendency (frequencies, means, medians, and standard deviations) described the socio-demographic characteristics of study participants (HIV-positive mother-baby pairs). A chi-square test of independence was used for the bivariate analysis of the relationships between outcomes and independent categorical variables. A p-value of 0.05 or less was used to determine the significance of the association between predictors and outcome variables. Then the multivariate analysis using multinomial logistic regression analysis was used to determine associations and ascertain the effect of confounders. The backward elimination method was used to determine the final model. First, the model contained all variables that showed significant association with the outcome variable during the bivariate analysis. The least significant variable (the one with the highest p-value) was removed from the model. The elimination continued until the stopping rule was reached. The stopping rule was when all remaining variables had a significant p-value (P = 0.05).

### Ethical consideration

Ethical clearance for the PMTCT project was obtained from the National Institution of Medical Research (NIMR) and the permission to use mother-baby pairs PMTCT Option B+ cascade data was received from the Ministry of Health, Community Development, Gender, Elderly, and Children, through the PMTCT coordination unit. Informed consent was waived because this was purely secondary data and hence, we did not interact with the subjects. The confidentiality of participants in the data was highly maintained, and special identification numbers were used to ensure anonymity.

## Results

### General characteristics

The mean age of mothers included in this analysis was 28.9 (SD = 5.8) years. The mean number of pregnancies (gravidity) was 3±2 and the mean gestation age at enrolment was 19.1±6.2 weeks. A total of 502 (78.9%) of 636 mothers who indicated their marital status were living with their partners (i.e. married or cohabited).

### Baseline HIV testing characteristics of mother-baby pairs

Of the 751 mother-baby records observed, 733 (97.6%, 95% CI: 96.2–98.5) had DBS1 collected by 2 months of age but only 332 (44.2%, 95% CI: 40.7–47.8) mother-baby pairs received the HIV confirmatory test within 18 months. Results of 651(86%, 95% CI: 84.0–89.0) mother-baby pairs who had DBS1 collected in the first two months were recorded, of which 641(98.46%, 95% CI: 97.2–99.3) were HIV-negative and only 10(1.54%, 95% CI: 0.7–02.8) were HIV positive. More than half of 429 (57.1%, 95% CI: 53.5–60.6) mothers were newly HIV positive and diagnosed upon enrolment; less than a quarter of 157 (20.9%, 95% CI: 18.1–24) mothers had tested partners; and the majority 644 (85.5%, 95% CI: 83.1–88.1) of mothers had treatment supporters (Table 1).

**Table 1. The baseline HIV testing characteristics of the 751 mother-baby pairs enrolled in the PMTCT cascade in selected health facilities between 2015 and 2017.**

| Variables | Frequency (%) | 95% Conf. Interv | | Confirmed HIV—Testing | | P-Value |
|---|---|---|---|---|---|---|
| | | | | No (%) | Yes (%) | |
| Maternal Age Group | Young (10–24) | 188 (25) | 22.1–28.2 | 120 (63.8) | 68 (36.2) | 0.010 |
| | Adults (25 –max) | 563 (75) | 71.7–77.9 | 299 (53.2) | 264 (46.9) | |
| | **Total** | **751 (100)** | | **419 (55.8)** | **332 (44.2)** | |
| Gravidity Groups | G1 –G2 | 286 (38.1) | 34.7–41.6 | 177 (61.9) | 109 (38.1) | 0.008 |
| | G3—Max | 465 (61.9) | 58.4–65.3 | 242 (52.0) | 223 (48.0) | |
| | **Total** | **751 (100)** | | **419 (55.8)** | **332 (44.2)** | |
| Gestation Age Group | 1st Trimester | 138 (18.4) | 15.8–21.3 | 87 (63.0) | 51 (37.0) | 0.058 |
| | 2nd & 3rd Trimester | 613 (81.6) | 78.7–84.2 | 332 (54.2) | 281 (45.8) | |
| | **Total** | **751 (100)** | | **419 (55.8)** | **332 (44.2)** | |
| Live with Partner | No | 134 (21.1) | 18.1–24.4 | 76 (56.7) | 58 (43.3) | 0.814 |
| | Yes | 502 (78.9) | 75.6–81.9 | 279 (55.6) | 223 (44.4) | |
| | **Total** | **636 (100)** | | **355 (55.8)** | **281 (44.2)** | |
| Maternal HIV status on Enrolment | Know HIV +ve | 322 (42.9) | 39.4–46.5 | 165 (51.2) | 157 (48.8) | 0.030 |
| | New HIV +ve | 429 (57.1) | 53.5–60.6 | 254 (59.2) | 175 (40.8) | |
| | **Total** | **751 (100)** | | **419 (55.8)** | **332 (44.2)** | |
| Partner Tested | No | 594 (79.1) | 76–81.8 | 347 (58.4) | 247 (41.6) | 0.005 |
| | Yes | 157 (20.9) | 18.1–24 | 72 (45.9) | 85 (54.1) | |
| | **Total** | **751 (100)** | | **419 (55.8)** | **332 (44.2)** | |
| DBS1 Collected | No | 18 (2.4) | 1.5–3.8 | 15 (83.3) | 3 (16.7) | 0.017 |
| | Yes | 733 (97.6) | 96.2–98.5 | 404 (55.1) | 329 (44.9) | |
| | **Total** | **751 (100)** | | **419 (55.8)** | **332 (44.2)** | |
| Treatment Supporter | No | 107 (14.3) | 11.9–16.9 | 70 (65.4) | 37 (34.6) | 0.030 |
| | Yes | 644 (85.7) | 83.1–88.1 | 349 (54.2) | 295 (45.8) | |
| | **Total** | **751 (100)** | | **419 (55.8)** | **332 (44.2)** | |

A chi-square test of independence was used to assess the relationship between independent variables and mother-baby pairs' HIV confirmatory testing within 18 months of age (confirmed HIV testing). There was a significant relationship between the following variables and confirmed HIV testing: maternal age group $X^2(1, N = 751) = 6.57$, p = 0.01; gravidity $X^2(1, N = 751) = 6.96$, p = 0.008; maternal HIV status at enrollment $X^2(1, N = 751) = 4.73$, p = 0.03; partner being tested $X^2(1, N = 751) = 7.94$, p = 0.005; collection dried blood spots for *HIV*-1 PCR tested $X^2(1, N = 751) = 5.67$, p = 0.02; and presence of treatment supporter tested $X^2(1, N = 751) = 4.69$, p = 0.03. There was not a significant relationship between the following variables and confirmed HIV testing: gestation age group at enrolment $X^2(1, N = 751) = 3.60$, p = 0.06 and marital status (Live with a partner) $X^2(1, N = 636) = 0.06$, p = 0.81. Table 1 below summarizes the results of the descriptive analysis of the baseline HIV testing characteristics of mother-baby pairs.

## Factors associated with confirmatory HIV testing among exposed children within 18 months of age

All factors that were statistically significantly associated with the dependent variable were entered into the regression analysis. The binary logistic regression was conducted to provide unadjusted odds ratios. Then multivariate logistic regression was used to obtain an adjusted odds ratio (OR). During the backward elimination method to determine the final model,

**Table 2. Logistic regression analysis of factors associated with turn up for confirmatory among HIV-positive mother-baby pairs who were enrolled in PMTCT cascade in selected health facilities between 2015 and 2017.**

| Factor | OR | P-Value | 95% CI | AOR | P-Value | 95% CI |
|---|---|---|---|---|---|---|
| Maternal Age Group | 1.56 | 0.011 | 1.11–2.19 | 1.44 | 0.040 | 1.02–2.04 |
| Gravidity Groups | 1.50 | 0.008 | 1.11–2.02 | 1.27 | 0.167 | 0.90–1.79 |
| Gestation Age Group | 1.4 | 0.058 | 0.99–2.11 | 1.43 | 0.075 | 0.97–2.11 |
| Partner tested | 1.66 | 0.005 | 1.16–2.36 | 1.74 | 0.003 | 1.21–2.50 |
| Maternal HIV Status | 0.72 | 0.030 | 0.54–0.97 | 0.72 | 0.033 | 0.53–0.97 |
| Treatment Supporter | 1.60 | 0.031 | 1.04–2.45 | 1.58 | 0.041 | 1.02–2.46 |
| DBS1 Collected | 4.07 | 0.027 | 1.17–14.18 | 3.61 | 0.048 | 1.01–12.84 |

gravidity and gestation age were not statistically significant and were eliminated consecutively (Table 2).

The results of multiple regression analysis of the factors included in the final model were: Maternal age group adults' mothers were 1.44 more likely to receive confirmatory HIV-testing for their babies than young mothers (Maternal age group AOR = 1.44 and p = 0.04). Mothers with partners who tested for HIV were 1.74 more times likely to have confirmatory HIV testing than those with partners not tested for HIV (Partner tested AOR = 1.74 and p = 0.003). New HIV-positive mothers diagnosed on enrolment were 0.72 less likely to turn up their babies for confirmatory HIV testing than known HIV-positive mothers (Maternal HIV status at enrolment AOR = 0.72 and p = 0.033); mothers with treatment supporters were 1.58 times more likely to receive confirmatory HIV-testing than mothers without treatment supporters (Treatment supporter AOR = 1.58 and p = 0.041), and finally, mothers with babies whose DBS1 was collected were 3.61 more likely to have confirmatory HIV-testing (DBS1 collection AOR = 3.61 and p = 0.048).

## Discussion

This study provides a picture of the implementation of Option B+ in Tanzania mainly focusing on early infant diagnosis (EID) of HIV among exposed infants. Confirmatory HIV testing is a very important part of EID to ensure that all affected infants are identified and enrolled in treatment. In this study 44.2% of mothers enrolled in Option B+ turned in their babies for confirmatory testing. The study revealed that: Maternal age—adult mothers were more likely to take their babies for confirmatory HIV testing than young mothers, and mothers with a tested partner were more likely to take their babies for confirmatory HIV testing than mothers with partners who have not been tested, newly diagnosed mothers were less likely to take their babies for confirmatory HIV testing than mothers who knew of their HIV status prior to the enrolment (Known HIV positive), mothers with treatment supporters were more likely to take their babies for confirmatory HIV testing than mothers without treatment supporters, and mothers who brought their babies for DBS-PCR-1 collection were more like to take for confirmatory HIV test within the 18-month post-delivery than those mothers who did not bring their babies for DBS-PCR-1 collection.

Lost to follow-up is one of the major problems that affect the uptake of confirmatory HIV testing [18]. The implementation of option B+ in Tanzania has remarkably increased the uptake of confirmatory HIV testing within 18 months of age. However, most mothers are still lost to follow-up after the first two months post-delivery. This study showed that 44% of mothers brought their babies for confirmatory HIV testing by end the 18 months period, which is higher than findings of past studies in Tanzania found only 34.6% of mothers brought their babies to the test [15, 19]. Despite the fact that DBS-CPR 1 collection has been associated with

the increase in the uptake of confirmatory HIV testing; still more than 50% of mothers who came for DBS-PCR 1 lost to follow-up by the end of 18 months of age. The finding is consistent with findings of past studies in Northern Tanzania, which revealed a high lost to follow-up after the first two months post-delivery as one of the setbacks in implementing Option B+ [19, 20]. Studies have associated the delay in the delivery of the DBS-PCR 1 results from the centralized laboratories used in Tanzania with lost to follow up. For instance, a recent study in six Sub-Saharan African countries has revealed that with centralized testing only 58% of caregivers accessed test results after 90 days [21]. This contributes to lost-to-follow-up and delay in starting ART for HIV-infected infants. The point-of-care early infant diagnosis (POC -EID) has shown potential to improve timely results and initiation of ART for HIV-infected children [21, 22]. However, more studies have to be done on the feasibility of its implementation in Tanzania. Likewise, it is not clear whether these mothers had enough knowledge their roles in the implementation of the Option B+ cascade of services until after 18 months. Probably they thought that the DBS-PCR 1 test results were confirmatory and there was no need for another test. Mothers of babies with HIV-negative DBS-PCR 1 results need intensified counseling and close follow-up just like those who test Positive. This will ensure that mother-baby pairs remain in treatment and babies remain HIV-free until they graduate at 18 months.

Being an adult mother and experienced with PMTCT has been found to increase compliance to Option B+. Adult mothers (aged 25 years and above) were more likely to turn up for confirmatory HIV tests at their babies 18 months of age HIV than young mothers (aged less than 25). The majority of the women in Tanzania have their first delivery in their youth age [23]. Hence, by the time they reach adulthood age (25 years), they have experienced reproductive health services and might have been exposed to PMTCT services thus knowing the importance of adherence to services. Likewise, mothers who were newly HIV diagnosed (less experienced with PMTCT) were less likely to turn up for confirmatory HIV test at the 18-month post-delivery than mothers who knew their HIV status before enrolment to antenatal care (experienced with PMTCT). Studies done in sub-Saharan Africa have indicated poor adherence to Option B+ among young and newly HIV-diagnosed mothers [3, 4]. Poor adherence among mothers enrolled in Option B+ have been found to be associated with a lack of experience with PMTCT services, short time to process and accept results, denial of the test results, fear of committing to a life-long treatment regimen, feeling of shame, and stigma [20, 24].

One of the unique findings in this study is that having a partner who has been tested for HIV is very important for compliance with Option B+ regardless of marital status. While marital status had no association with mothers turning in their babies for confirmatory HIV testing; mothers who had their partner tested for HIV are more likely to turn up for confirmatory HIV tests at their babies 18 months of age. This is contrary to recent studies in sub-Saharan Africa, which revealed that married mothers were more likely to adhere to option B+ than unmarried mothers [25, 26]. Awareness of the partner's HIV status has been associated with compliance with Option B+ services. Studies in Malawi, Ethiopia, and Kenya revealed that mothers who know the HIV status of their partners are more likely to remain in Option B + care than those who did not know [27]. The finding from this study goes the extra mile as it indicates mothers who know that their partners have tested for HIV regardless of knowledge of their HIV status have an impact on the compliance to Option B+ care. Unfortunately, in this study, less than a quarter of mothers involved in the analysis had their partners tested. The calls for more effort to educate male partners on the importance of HIV testing.

The presence of treatment supporters improves mothers' compliance to Option B+ cascade of services. The majority of mothers involved in this study had a treatment supporter. These findings support studies showing that treatment supporters increase compliance with Option

B+ [20, 27–29]. A treatment supporter can be a couple, a sibling, a relative, or a friend who a mother has selected to assist. The use of the treatment supporters has been a successful technique used in patients with chronic conditions that require long-term course treatment. For instance, treatment supporters have been useful for many years in Tanzania to ensure patients' adherence to the tuberculosis treatment regime [30, 31]. The study conducted Kilimanjaro region on the predictors of postpartum HIV care engagement among women in the PMTCT program, suggested strengthening the social support network system by engaging treatment supporters in HIV care [17, 20, 24, 32]. Therefore, findings from this study cement the importance of engaging treatment supporters not only in engaging mothers to care but also to ensure that their babies receive the necessary HIV cares.

### Limitation

This study-employed data from the ministry of health management information system (HMIS) and the PMTCT cascade survey data collected from 80 health facilities. The following limitations were highlighted in this study: Data quality: only 751 of 4738 data collected were complete. This might have affected or influenced some of the results. There is a need to train health workers on the importance of proper documentation, and constant supervision, and employ more health workers to cover the shortage. Likewise, these findings are from the analysis of secondary data that were primarily collected from the randomly selected health facilities for routine monitoring activities of PMTCT services. Therefore, it is difficult to know exactly how well the data collection process was done.

### Conclusion

This study has found that the turn-up for confirmatory HIV testing within 18 months among mother-baby pairs enrolled in the Option B+ approach is still low in Tanzania. More than half of mothers lost to follow-up after the first two months post-delivery. This low turn-up for the confirmatory HIV testing among mother-baby pairs enrolled in Option B+ is associated with Low maternal age (young mothers), the low uptake of HIV testing among male partners (having a male partner who has not tested for HIV), being a newly diagnosed HIV mother (lack of experience to HIV services), lack of treatment supporters, and failure to take the first DBS-PCR HIV test within the first two months post-delivery. These factors have to be addressed in the Option B+ implementation guidelines in order to increase the retention of mother-baby pairs into the Option+ B cascade of services.

### Recommendations for future directions

The study recommends the following: Post-HIV testing counseling sessions among young mothers (mothers under 25 years of age) and newly HIV-diagnosed mothers be intensified; male partner involvement in reproductive services, couple counseling, and the importance of HIV result disclosure should be enhanced; and further studies be done to identify the best modalities of identifying, integrating and utilizing treatment supporters who will assist HIV infected mother to comply Option B+ cascade.

### Author Contributions

**Conceptualization:** Baraka M. Morris, Edith Tarimo, Bruno Sunguya.

**Data curation:** Mukome Nyamhagata.

**Formal analysis:** Baraka M. Morris, Edith Tarimo, Bruno Sunguya.

**Funding acquisition:** Baraka M. Morris, Edith Tarimo.

**Investigation:** Baraka M. Morris.

**Methodology:** Baraka M. Morris, Bruno Sunguya.

**Project administration:** Baraka M. Morris.

**Resources:** Baraka M. Morris, Mukome Nyamhagata.

**Software:** Baraka M. Morris.

**Supervision:** Mukome Nyamhagata, Edith Tarimo, Bruno Sunguya.

**Validation:** Baraka M. Morris, Mukome Nyamhagata, Edith Tarimo.

**Visualization:** Baraka M. Morris, Mukome Nyamhagata.

**Writing – original draft:** Baraka M. Morris.

**Writing – review & editing:** Baraka M. Morris, Edith Tarimo, Bruno Sunguya.

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
