## [Decision Letter · Decision Letter 0]

8 Nov 2022

PGPH-D-22-01598

Eliminating mother-to-child transmission of HIV in Tanzania calls for efforts to address factors associated with a low confirmatory test

Dear Dr. Morris,

Thank you for submitting your manuscript to PLOS Global Public Health. After careful consideration, we feel that it has merit but does not fully meet PLOS Global Public Health’s publication criteria as it currently stands. Therefore, we invite you to submit a revised version of the manuscript that addresses the points raised during the review process.

We look forward to receiving your revised manuscript.

Kind regards,

Abram L. Wagner, PhD, MPH

Academic Editor

Journal Requirements:

a. Please clarify all sources of funding (financial or material support) for your study. List the grants (with grant number) or organizations (with url) that supported your study, including funding received from your institution. 

b. State the initials, alongside each funding source, of each author to receive each grant.

2. Please indicate by return email the full and correct funding information for your study and confirm the order in which funding contributions should appear.

Additional Editor Comments (if provided):

Reviewers' comments:

Reviewer's Responses to Questions

**Comments to the Author**

1. Does this manuscript meet PLOS Global Public Health’s publication criteria? Is the manuscript technically sound, and do the data support the conclusions? The manuscript must describe methodologically and ethically rigorous research with conclusions that are appropriately drawn based on the data presented.

Reviewer #1: Yes

Reviewer #2: Yes

2. Has the statistical analysis been performed appropriately and rigorously?

Reviewer #1: Yes

Reviewer #2: Yes

3. Have the authors made all data underlying the findings in their manuscript fully available (please refer to the Data Availability Statement at the start of the manuscript PDF file)?

Reviewer #1: Yes

Reviewer #2: Yes

4. Is the manuscript presented in an intelligible fashion and written in standard English?

Reviewer #1: Yes

Reviewer #2: Yes

5. Review Comments to the Author

Reviewer #1: Overview: The manuscript contributes uniquely to the literature and delineates information on PMTCT -option B+ cascade in Tanzania. The potential factors in the turning up of mothers-baby pairs, especially regarding DBS-PCR1

Organization of Article: The article is well-written and summarizes the factors associated with the turning-up of mother-baby pairs on PMTCT -option B+. The research aim for the study was well-defined, but it seems to focus on aspects related to confirmatory HIV testing. The author should discuss the treatment adherence level of PMTCT elaborately among mother-baby.

Study Design: This would probably have benefitted from a validated instrument to assess the factors and adherence level. With data being secondary, some vital information might have been omitted.

Literature Review: The various article is current and reflects issues found on PMTCT -option B+. Since PMTCT adherence varies, the Author should include current information on Option B mother-child adherence to treatment level.

Suggested Revision

Authors need to include how data will be protected after the study.

This article would benefit from information on Antenatal data.

The study objective implies monitoring and evaluating treatment among mother-baby enrolled in the option B approach of PMTCT, more discussion of this aspect would be beneficial.

Most of the discussion focuses on factors related to option B compliance, with little discussion on the loss to follow up.

Recommendations: Accept with Revisions

Reviewer #2: This paper can be improved as follow:

1. Overall the study stated the early infant diagnosis, however the results of EID did not exist (information only whether DBS is collected or not), it will be better if adding information and comparing between those with EID Positive and EID negative results with other factors associated with it.

2. New WHO guidelines inform that in Page 5: WHO recommends that all HIV-exposed infants receive a virological test for HIV within six weeks of birth, followed by the initiation of ART for those who are infected;

source : https://www.unicef.org/supply/sites/unicef.org.supply/files/2020-03/HIV-early-infant-diagnosis-viral-load-point-care-diagnostics-market-note.pdf

3. Line 54: adding reference

4. Line 208: deleted the last variable as already put in the column

5. Line 232-234: table need to be revise and add the information stated in line 216-218

6. Recommendation that needs to be added is based on WHO guidelines is early detection of infants using DBS within six weeks of birth and directly provided treatment once infected.

7. Below similar journal that can be cited to enrich the findings :

a. Article Source: Interventions to increase early infant diagnosis of HIV infection: A systematic review and meta-analysis

Okusanya B, Kimaru LJ, Mantina N, Gerald LB, Pettygrove S, et al. (2022) Interventions to increase early infant diagnosis of HIV infection: A systematic review and meta-analysis. PLOS ONE 17(2): e0258863. https://doi.org/10.1371/journal.pone.0258863

b. Point-of-care testing can achieve same-day diagnosis for infants and rapid ART initiation: results from government programmes across six African countries https://doi.org/10.1002/jia2.25677

access through https://onlinelibrary.wiley.com/doi/10.1002/jia2.25677

6. PLOS authors have the option to publish the peer review history of their article (what does this mean?). If published, this will include your full peer review and any attached files.

**Do you want your identity to be public for this peer review?** For information about this choice, including consent withdrawal, please see our Privacy Policy.

Reviewer #1: No

Reviewer #2: No

---

## [Editor Report · Decision Letter 1]

29 Dec 2022

Eliminating mother-to-child transmission of HIV in Tanzania calls for efforts to address factors associated with a low confirmatory test

PGPH-D-22-01598R1

Dear mr. Morris,

We are pleased to inform you that your manuscript 'Eliminating mother-to-child transmission of HIV in Tanzania calls for efforts to address factors associated with a low confirmatory test' has been provisionally accepted for publication in PLOS Global Public Health.

Best regards,

Abram L. Wagner, PhD, MPH

Academic Editor